# Prevalence, severity and associated risk factors of anemia among human immunodeficiency virus-infected adults in Sawla General Hospital, Southern Ethiopia: A facility-based cross-sectional study

**Rishan Hadgu, Ahmed Husen, Esayas Milkiyas, Niguse Alemayoh, Robel Zemoy, Azene Tesfaye, Dagimawie Tadesse, Aseer Manilal, Aklilu Alemayehu** *

Department of Medical Laboratory Sciences, College of Medicine and Health Sciences, Arba Minch University, Arba Minch, Ethiopia

* aaakealex59@gmail.com

## Abstract

### Background

Anemia is a significant public health problem in HIV/AIDS patients worldwide. This study is aimed to determine the prevalence of anemia and its risk factors among HIV-infected adults in Sawla General Hospital, southern Ethiopia.

### Methods

A facility-based cross-sectional study involving HIV-infected adults was conducted in ART clinic of Sawla General Hospital from April 01 to May 31, 2019. A systematic random sampling technique was employed to recruit the study participants. Socio-demographic and clinical data were collected using a structured questionnaire and checklist. Hemoglobin concentration from venous blood was determined by HemoCue® 301 analyzer. Descriptive and inferential statistics, by Statistical Package for Social Science version 26.0, were applied; p-values $\leq 0.05$ in the multivariable logistic regression analysis were considered statistically significant.

### Results

A total of 220 HIV-infected adults participated in this study. The prevalence of anemia was 38.6%, from which 90.6, 7.1, and 2.3% are mild, moderate, and severe anemia, respectively. Anemia among HIV-infected adults was significantly associated with $CD_4$ cell count below 200 cells/mm$^3$ (AOR: 4.32; 95% CI: 2.10–8.86), clinical stage III or above (AOR: 4.20; 95% CI: 1.06–16.62), five or more years duration of HIV infection (AOR: 2.32; 95% CI: 1.08–4.94) and BMI below 18.5 kg/m$^2$ (AOR: 3.82; 95% CI: 1.83–8.00).

**Data Availability Statement:** The data used for this study are uploaded as supportive information in zipped file as "S1 Dataset".

**Funding:** The authors received no specific funding for this work.

**Competing interests:** The authors have declared that no competing interests exist.

**Abbreviations:** AIDS, Acquired Immunodeficiency Syndrome; AOR, Adjusted Odds Ratio; ART, Anti-retroviral Therapy; BMI, Body Mass Index; $CD_4$, Cluster of Differentiation; CI, Confidence Interval; COR, Crude Odds Ratio; EDTA, Ethylene Diamine Tetra Acetic Acid; ETB, Ethiopian Birr; g/dL, Gram per Deciliter; HAART, Highly Active Anti-retroviral Therapy; HIV, Human Immunodeficiency Virus; SGH, Sawla General Hospital; VL, Viral Load.

## Conclusion

Anemia is a moderate public health problem among the study population. Longer duration of HIV infection, advanced clinical stage, lower $CD_4$ cell count, and BMI are risk factors for anemia. Therefore, early ART enrolment for HIV-infected adults with nutritional support and rigorous monitoring of $CD_4$ cell count are essential to lower the prevalence.

## Background

Anemia is a condition characterized by inadequacy of the number of red blood cells (RBCs) and/or their oxygen-carrying capacity to meet the physiologic needs of the body [1, 2]. It is often determined by measuring hemoglobin (Hb) concentration in whole blood. The cutoff value of Hb concentration to determine anemia varies by sex, age, altitude, pregnancy, and smoking status of an individual. Moreover, the Hb concentration of an individual and prevalence of anemia form the basis for classifying the severity of anemia as mild, moderate, and severe problem [1].

Anemia in HIV-infected individuals can result from decreased production or increased destruction of RBCs [2–4]. The pathophysiological mechanisms include blood loss due to neoplastic diseases, decreased RBC production, increased hemolysis, and ineffective RBC production due to nutritional deficiency or bone marrow suppression caused by the virus or certain medications [1, 3, 5].

According to the guideline of comprehensive HIV/AIDS management, ART should be initiated early for all HIV-positive adults irrespective of their WHO clinical stages and CD4 cell count [6, 7]. Identifying the underlying cause is crucial in managing anemic HIV-infected persons. Monitoring Hb level simultaneously with VL is recommended [5, 8, 9]. Therapeutic options such as HAART, epoetin, blood transfusion, and other hematinic may be recommended based on the severity of anemia [5].

Anemia among HIV-infected patients is an important health problem in the world [10]. In a recent systematic review and meta-analysis, its pooled prevalence was 46.6% among HIV-infected adults worldwide [11]. Studies conducted in China, Nigeria, and South Africa have reported a prevalence of anemia among HIV patients ranging from 25.8% to 75% [12–15]. Ethiopia carries high burden of anemia, where 31.0% pooled prevalence is found among HIV-infected adults [16]. Previous research done in three cities of Ethiopia reported a prevalence ranging from 23 to 41.3% [17–20].

Various factors affect the risk of acquiring anemia among HIV patients [3, 5, 16]. Notably, variables such as the patient's sex, zidovudine treatment, low CD4 cell count, high viral load, and lower body mass index may contribute to a higher likelihood of anemia in HIV/AIDS patients [15, 16, 18, 20–22].

Anemia is a significant hematologic disorder among HIV-infected individuals [5]. It can result in impaired physiological function, decreased quality of life, reduced survival, and increased disease progression in HIV/AIDS patients [5, 9]. Employment problems and sleep disturbances are among the factors that contribute to morbidity and disability in anemic HIV-positive patients, making it a crucial public health problem [5].

Understanding the burden, mechanism, and associated factors of anemia in HIV-infected people is essential to designing interventions for reducing morbidity and mortality from HIV/AIDS. However, the prevalence of anemia and its associated factors among adult HIV patients on HAART are not well documented in the current study area, and this study is aimed to address this gap in Sawla General Hospital (SGH), southern Ethiopia.

## Method and materials

### Study setting and design

A facility-based cross-sectional study was conducted from April 01 to May 31, 2019, in the ART clinic of SGH. An SGH is a public hospital located in Sawla Town, which is the capital of the Gofa zone in the Southern Nations, nationalities, and peoples' Region of Ethiopia (SNNPR). Sawla Town is located 515 km to the south of Addis Ababa, the nation's capital, and 285 km to the southwest of Hawassa, the capital of SNNPR. An SGH serves about one million population in the zone and its surrounding. The hospital has been providing ART service for over 1000 HIV-infected clients since its establishment in 2000 [23].

### Population, eligibility, sample size, and sampling technique

The source population was all HIV-infected adults attending the ART clinic of SGH. HIV-positive patients aged 15 to 64 years who are on ART for at least six months are included in this study. Pregnant women; those who received a blood transfusion within three months before the study period; those on an iron supplement, and severely ill patients were excluded from the current study.

The total sample size was determined by using a single population proportion formula using 23.0% prevalence from a previous similar study conducted in Debre-Tabor Hospital, northwest Ethiopia [18]. The initial sample size was 272, but after adjusting for population correction and 10% non-response rate consideration, the final sample size became 220 (Eq 1).

$$n = \frac{(Z_{\alpha/2})^2 * P * (1 - P)}{d^2}$$

Eq 1: **Formula to calculate sample size** [24].

Where, n: sample size, Z: statistic for confidence level (1.96 at 95% confidence level), P: prevalence = 23.0%, d: margin of error = 0.05.

A systematic random sampling technique was used to select 220 participants among 830 HIV-infected adults (K = 4), who consulted the ART clinic during the study period and met the inclusion criteria.

### Study variables

Anemia was the dependent variable for this study. Age, sex, education, occupation, monthly income, residence, WHO clinical stage of HIV/AIDS, $CD_4$ cell count, VL, body mass index (BMI), drug regimen type, duration of HIV infection, and HAART, were taken as independent variables.

### Data collection tool and procedure

Data on socio-demographic and anthropometric variables (age, sex, residence, educational level, occupation, monthly income, and BMI) were collected by using a pretested and structured questionnaire. Data on clinical (HIV/AIDS WHO clinical stage, duration of HIV infection, drug regimen and HAART exposure) and immunological ($CD_4$ cell count, and VL) variables were collected from the patient's records using a checklist. The questionnaire and the checklist were prepared after reviewing different related literature. The questionnaire was prepared in English, translated into Amharic, and translated back to English to check for its consistency. Then, this questionnaire was pretested in Arba Minch General Hospital by taking 5% of the total sample size. Based on the pretest result, we did not made major modification to the questionnaire since we did not find major gap on it.

Weight was measured in kilograms (kg) while the study subject is wearing no heavy clothes or shoes. Height was measured using a stadiometer to the nearest 0.5 centimeters while the study subject is wearing no shoes, cap, or headgear and standing with the back to the measuring rod, and looking straight ahead. Then, BMI was calculated as weight in kg divided by height in meters square.

Venous blood sample was collected from HIV-infected adult, who gave consent. A standard clean venipuncture technique was used to collect 3 ml of venous blood from each subject from the median cubital vein using an EDTA-anticoagulated tube. From the collected blood, Hb concentration was determined by using HemoCue 301 Analyzer. The HemoCue Hb determination system works based on oxidation of Hb to methemoglobin. The system involves drawing undiluted approximately 10 uL capillary or venous whole blood into a chemically coated single-use micro-cuvette by capillary action followed by lysis of RBCs by sodium deoxycholate releasing the Hb that will react with sodium nitrite and sodium azide resulting in azidemethemoglobin. The absorbance of the azidemethemoglobin is measured by HemoCue photometer at wavelengths of 506 nm and 880 nm to compensate for possible turbidity. Then, the Hb concentration is digitally displayed in g/dL within 30 seconds [25]. To determine anemia, the obtained Hb concentration was further characterized according to the 2011 WHO guideline [1].

## Data processing and analysis

All data from the questionnaire, laboratory results, and medical records were checked manually for completeness and clarity before proceeding to any data analysis. Data were entered into Statistical Package for Social Sciences (SPSS) software Version 26.0. Descriptive statistics such as frequency, mean, median and range were calculated to describe the study participants. Bi-variable and multivariable logistic regression models were used to identify independent predictors of anemia. Those with P-values $\leq 0.2$ in the bi-variable logistic regression analysis were taken into multivariable logistic regression analysis. Hosmer and Lemeshow's goodness of fit for the logistic regression test was used to check model fitness, and the fitness was confirmed by $P > 0.05$. Finally, P-value $\leq 0.05$ in the multivariable logistic regression analysis was considered a statistically significant association between independent variables and anemia [26].

## Data quality control

Data quality was ensured from collection to the final identification step in the laboratory by following the prepared standard operating procedures. Pre-analytical quality was assured by employing an in-house SOP for venous blood collection. Blood sample analysis was done within eight hours of collection. The questionnaire was pretested, training was given to data collectors, and close supervision was done during data collection by supervisor. All completed questionnaires were daily assessed for their completeness, clarity, and consistency during the data collection period by supervisor. Finally, careful recording and interpretation of the result were made during the post-analytical quality assurance phase.

## Operational definition

**HIV-infected adult.** An HIV-positive person aged 15–64 years, who is on ART register of SGH [8].

**Anemia.** Hb concentration less than 12.0 g/dL for women and 13.0 g/dL for men [1].

**Mild anemia.** Hb concentration 11.0–12.9 g/dL for men and 11.0–11.9 g/dL for women [1].

**Moderate anemia.** Hb concentration 8.0–10.9 g/dL for both men and women [1].

**Severe anemia.** Hb concentration below 8.0 g/dL for both men and women [1].

**Underweight adults.** Adults with BMI below 18.5 kg/m$^2$ [1].

**Overweight adults.** Adults with BMI 25.0–29.9 kg/m$^2$ [1].

**First line regimen.** TDF + 3TC + EFV [8].

**Second line regimen.** AZT + 3TC + LPV/r or ATV/r [8].

## Results

### Socio-demographic and anthropometric characteristics of study participants

A total of 220 adult HIV-positive participants were included in this study. The mean age of the study participants was 34.15 ± 10.78, with minimum and maximum ages of 15 and 63 years, respectively. The majority of the study participants are females (51.4%), urban residents (61.4%), and government employees (40.0%). The average monthly income of the participants was 1695.00 ± 1557.47 ETB. The average BMI is 21.00 ± 4.08 kg/m$^2$, with minimum and maximum BMI of 15.26 kg/m$^2$ and 40.34 kg/m$^2$, respectively (Table 1).

### Clinical and immunological characteristics of the participants

Above three-quarters of the participants are on the WHO HIV/AIDS clinical stage I and taking first-line ART drugs. The mean CD$_4$ cell count was 408.15 ± 322.50 cells/mm$^3$, with the minimum and maximum, 45 and 1530 cells/mm$^3$, respectively. The minimum and maximum VL were 145 of 14726 copies/ml, respectively, and eighteen patients had an undetectable level. The majority (87.3%) of the participants were not taking cotrimoxazole prophylaxis. The mean duration of HIV infection was 55.38 ± 27.52 months, and the minimum and maximum were six and 132 months, respectively. The mean duration of HAART exposure was 42.30 ± 20.67 months, and the minimum and maximum were six and 90 months, respectively (Table 2).

### Overall prevalence and severity of anemia among HIV-infected adults

The mean Hb concentration was 12.99 ± 1.47 g/dL with minimum and maximum results of 7.80 g/dL and 15.70 g/dL, respectively. The mean Hb concentration of females and males was 12.88 ± 1.60 g/dL and 13.22 ± 1.19 g/dL, respectively. The overall prevalence of anemia among HIV-infected adults was 38.6% (95% CI: 32.2, 45.4%). The prevalence is 40.2% among men and 37.2% among women. Of 85 anemic patients, 77, 6, and 2 have mild, moderate, and severe anemia, respectively (Fig 1).

**Risk factors associated with anemia among HIV-infected adults.** Of the total of 220 study participants, above two-third of participants using a second-line drug regimen, and about a third of those found in WHO clinical stage II or above, respectively were anemic. More than half of the participants with at least five years of HIV infection and HAART use were anemic. Also, about half of those with a BMI below 18.5 kg/m$^2$ and those exposed to cotrimoxazole were anemic. Above 40% of male and old age patients as well as those lacking formal education and job were anemic. But some of these differences were not statistically significant.

Multivariable logistic regression analysis was done for the variables with P value ≤ 0.20 in binary logistic regression analysis to identify risk factors associated with anemia among HIV-infected adults. Based on the cutoff point, eight variables were taken as candidates for the multivariable logistic regression analysis. Finally, anemia among HIV-infected adults was significantly associated with being HIV-infected for five or more years (AOR: 2.32; 95% CI: 1.08–4.94, P = 0.030); being on clinical stage III or more (AOR: 4.20; 95% CI: 1.06–16.62, P = 0.041);

**Table 1. Socio-demographic and anthropometric characteristics of study participants at SGH, Southern Ethiopia, 2019 (n = 220).**

| Variables | Frequency | Percentage (%) |
|---|---|---|
| **Sex** | | |
| Female | 113 | 51.4 |
| Male | 107 | 48.6 |
| **Age (Years)** | | |
| 15–24 | 40 | 18.2 |
| 25–34 | 86 | 39.1 |
| 35–44 | 60 | 27.3 |
| 45–54 | 17 | 7.7 |
| 55–64 | 17 | 7.7 |
| **Residence** | | |
| Urban | 135 | 61.4 |
| Rural | 85 | 38.6 |
| **Education** | | |
| Unable to read and write | 41 | 18.6 |
| Primary school | 93 | 42.3 |
| Secondary school | 45 | 20.5 |
| Higher education | 41 | 18.6 |
| **Occupation** | | |
| Governmental | 88 | 40.0 |
| Housewife | 23 | 10.5 |
| Merchant | 64 | 29.1 |
| Student | 13 | 5.9 |
| Jobless | 27 | 12.3 |
| Other jobs | 5 | 2.3 |
| **Monthly income (ETB)** | | |
| 1–1499 | 131 | 59.6 |
| 1500–2999 | 70 | 32.7 |
| $\geq$ 3000 | 19 | 7.7 |
| **BMI (kg/m$^2$)** | | |
| < 18.5 | 78 | 35.5 |
| 18.5–24.9 | 106 | 48.2 |
| $\geq$ 25.0 | 36 | 16.4 |

having $CD_4$ cell count below 200 cells/mm$^3$ (AOR: 4.32; 95% CI: 2.10–8.86, P< 0.001) and BMI below 18.5 kg/m$^2$ (AOR: 3.82; 95% CI: 1.83–8.00, P< 0.001) (Table 3).

## Discussion

Anemia is a daunting public health problem among HIV-infected people and it involves complex pathogenesis. Although its etiology among HIV-infected adults is multifactorial, the burden is mainly attributable to anemia of chronic disease, viral suppression of hemopoiesis, opportunistic infections, nutritional deficiency and ART drugs [2, 7]. This study was aimed to determine the prevalence of anemia and its associated risk factors among HIV-infected adults in SGH. The overall prevalence of anemia was 38.6% (95% CI: 32.2, 45.4%). Of those 85 anemic patients, 90.6% have mild, 7.1% moderate, and 2.3% severe anemia. The prevalence was 40.2% among men and 37.2% among women. The prevalence of anemia among this

**Table 2. Clinical and immunological characteristics of study participants at SGH, Southern Ethiopia, 2019 (n = 220).**

| Variable | Frequency | Percentage (%) |
|---|---|---|
| **$CD_4$ cell count (cells/mm$^3$)** | | |
| < 200 | 92 | 41.8 |
| ≥ 200 | 128 | 58.2 |
| **Viral load (Copy/ml)** | | |
| < 1000 | 105 | 47.7 |
| ≥ 1000 | 115 | 52.3 |
| **Drug regimen** | | |
| First line | 171 | 77.7 |
| Second line | 49 | 22.3 |
| **WHO HIV stage** | | |
| Stage I | 166 | 75.5 |
| Stage II | 29 | 13.2 |
| Stage III/IV | 25 | 11.3 |
| **Duration of HIV infection (months)** | | |
| < 60 | 124 | 56.4 |
| ≥ 60 | 96 | 43.6 |
| **Duration of HAART exposure (months)** | | |
| < 60 | 149 | 67.7 |
| ≥ 60 | 71 | 32.3 |
| **Exposure to cotrimoxazole** | | |
| Exposed | 28 | 12.7 |
| Not exposed | 192 | 87.3 |

population is high, which can potentially hinder the treatment outcome and hence increase morbidity and mortality.

This finding is consistent with results from a systematic review and meta-analysis done in Ethiopia (31.0%) [16], as well as from a series of studies done in different areas of Ethiopia,

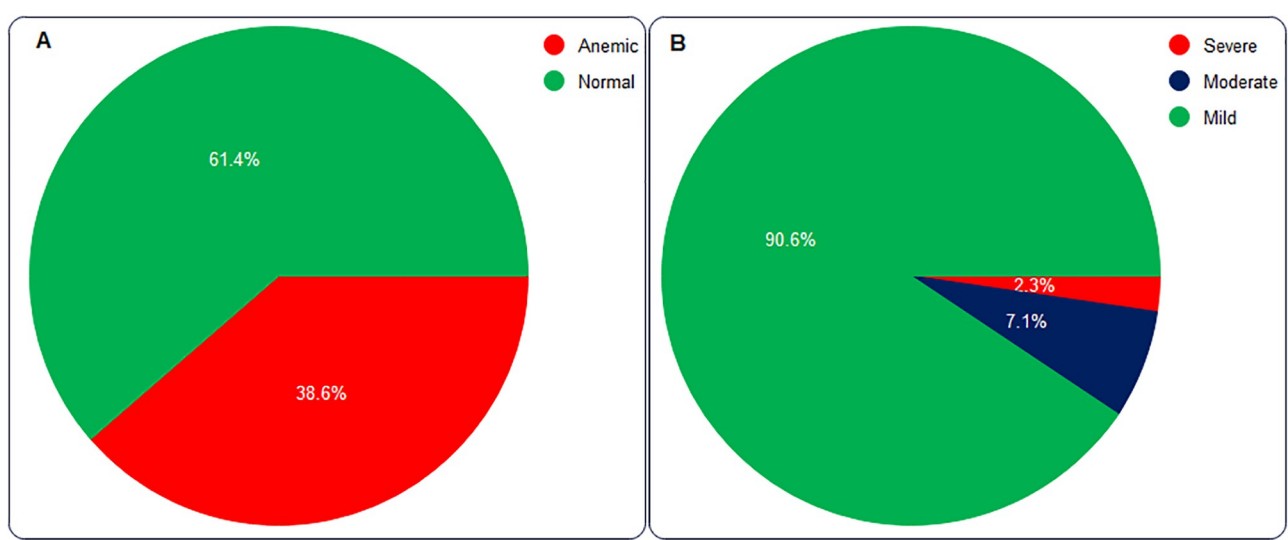

**Fig 1. Prevalence of anemia.** (**A**) Overall prevalence of anemia (n = 220), and (**B**) Severity of anemia (n = 85) among HIV-infected adults in SGH, southern Ethiopia, 2019.

**Table 3. Associated risk factors and distribution of anemia by characteristics of participants at SGH Southern Ethiopia, 2019 (n = 220).**

| Variables | Anemia | | COR (95% CI) | P-value | AOR (95% CI) | P-value |
|---|---|---|---|---|---|---|
| | Yes (%) | No (%) | | | | |
| **Residence** | | | | | | |
| Rural | 38(44.7) | 47(53.3) | 1.51(0.87,2.64) | 0.143 | 1.61(0.83,3.14) | 0.160 |
| Urban | 47(34.8) | 88(65.2) | 1 | | 1 | |
| **CD$_4$ cell count (cells/mm$^3$)** | | | | | | |
| < 200 | 55(59.8) | 37(40.2) | 5.31(2.94,9.56) | 0.000 | 4.32(2.10,8.86) | 0.000* |
| ≥ 200 | 30(23.4) | 98(76.6) | 1 | | 1 | |
| **Viral load (copies/ml)** | | | | | | |
| < 1000 | 30(28.6) | 75(71.4) | 1 | | 1 | |
| ≥ 1000 | 55(47.8) | 60(52.2) | 2.41(1.37,4.23) | 0.002 | 0.85(0.40,1.79) | 0.674 |
| **Drug regimen** | | | | | | |
| First line | 62(36.3) | 109(63.7) | 1.72(0.84,3.50) | 0.135 | 2.22(0.81,6.06) | 0.118 |
| Second line | 26(66.67) | 13(32.23) | 1 | | 1 | |
| **WHO HIV/AIDS clinical stage** | | | | | | |
| Stage I | 56(33.7) | 110(66.3) | 1 | | 1 | |
| Stage II | 12(41.4) | 17(58.6) | 1.38(0.62,3.10) | 0.427 | 1.75(0.63,4.84) | 0.282 |
| Stage III/IV | 8(32.0) | 15(68.0) | 4.17(1.69,10.26) | 0.002 | 4.20(1.06,16.62) | 0.041* |
| **BMI (kg/m$^2$)** | | | | | | |
| < 18.5 | 47(60.2) | 31(39.8) | 4.91(2.59,9.29) | 0.000 | 3.82(1.83,8.00) | 0.000* |
| 18.5–24.9 | 25(23.6) | 81(76.4) | 1 | | 1 | |
| ≥ 25.0 | 23(58.9) | 13(41.1) | 1.83(0.81,4.01) | 0.145 | 1.01(0.37,2.71) | 0.968 |
| **Duration of HIV infection (months)** | | | | | | |
| < 60 | 31(25.0) | 93(75.0) | 1 | | 1 | |
| ≥ 60 | 54(56.2) | 42(43.8) | 4.20(2.36,7.484) | 0.000 | 2.32(1.08,4.94) | 0.030* |
| **Exposure to cotrimoxazole** | | | | | | |
| Exposed | 14(50.0) | 14(50.0) | 1 | | 1 | |
| Not exposed | 71(37.0) | 121(63.0) | 0.58(0.26,1.30) | 0.190 | 1.14(0.38,3.37) | 0.809 |

*: Statistically significant

such as Wolaita Sodo (36.5%) [20], northeast Ethiopia (41.3%) [17], Gedeo Zone (34.8%) [26], and also in China (39.2%) [27].

The prevalence in our study setting is higher than reported from Debre-Tabor Hospital, northwest Ethiopia (23.0%) [18], southwest Ethiopia (23.0%) [19], southwest China (27.5%) [12], and the USA (7.18%) [21]. The lower socio-economic status of participants in our study than those in southwest China and the USA might have contributed to the observed disparity [12, 21]. Inclusion of a relatively larger number of HIV-infected adults with a CD$_4$ cell count < 200 cells/mm$^3$ and HAART-users than those studies done in Debre-Tabor Hospital and Jimma University Specialized Hospital, respectively, might have raised the prevalence in our setting.

Our finding is lower than the results of several studies [13, 14, 28, 29]. Studies conducted in Calabar, Nigeria (76.0%) [14], Southeastern Nigeria (76.3%) [13], Uganda (47.8%) [28], and China (51.9%) [29] reported higher prevalence than ours. The inclusion of many female participants in those studies done in Nigeria could be one possible reason for the disparity. Furthermore, our finding is also lower than the prevalence reported in China (51.9%). This is because the research in China included only newly diagnosed HIV patients, unlike to ours, which included participants exposed to HAART [29].

Various factors determine the risk of anemia among HIV-infected adults in multidimensional ways [3, 4]. In the current study, HIV-infected adults with a $CD_4$ cells count below 200 cells/mm$^3$, BMI below 18.5 kg/m$^2$, clinical stage III or above, and having HIV infection for five or more years have increased the risk of anemia.

Anemia among HIV-infected adults was statistically significantly associated with being HIV-positive for five or more years (AOR: 2.32; 95% CI: 1.08–4.94, P = 0.030). Similarly, individuals living with HIV for five to eight years had an increased likelihood of anemia by 2.59-fold (95% CI: 1.02–6.57) in a study conducted in Wolaita Sodo, southern Ethiopia [20]. The longer the patient stays with the virus, the higher the risk of anemia since the advancement brings about bone marrow suppression and heightened hemolytic tendency [3, 5].

In HIV-infected adults, having a $CD_4$ cell count below 200 cells/mm$^3$ raised the risk of anemia by 4.32 (95% CI: 2.10–8.86, P< 0.001). In line with our finding, having a $CD_4$ cells count below 200 cells/mm$^3$ has increased the risk of anemia by 2.15-fold (95% CI: 1.21–3.82) and 4.2-fold (95% CI: 2.03–8.67) in Wolaita Sodo [20] and Zewditu Memorial Hospital [22], respectively. Similar findings were reported from studies done in southeastern Nigeria [13] and Ethiopia [16]. The ability of HIV infection to suppress hemopoiesis might explain the observed association [2, 3].

Anemia was more than fourfold more likely in HIV-infected adults who were in clinical stage III or higher (AOR: 4.20; 95% CI: 1.06–16.62, P = 0.041). Likewise, patients at clinical stage III/IV had a twice (AOR: 2.03; 95% CI: 1.45–2.83) higher chance of anemia in a study done in Zewditu Memorial Hospital, Ethiopia [22] and by a 2.5-fold increased risk in a systematic review and meta-analysis done in Ethiopia [16]. It is plausible that as the disease advances, the risk of anemia could rise through intensified hemolytic tendency and suppressed production [3].

In this study, HIV-infected adults with a BMI below 18.5 kg/m$^2$ were nearly four-fold (AOR: 3.82; 95% CI: 1.83–8.00, P< 0.001) more likely to be anemic than those with an average body weight. Our finding is concordant with the result from Wolaita Sodo, where HIV patients with a BMI below 18.5 kg/m$^2$ had by an almost threefold (AOR: 2.96; 95% CI: 1.37–6.39) increased risk of anemia [20]. Furthermore, similar findings were reported from other parts of Ethiopia [16]. The observed association is biologically plausible since being underweight can indicate a nutritional deficiency, including iron, folate, and cobalamin, that might underpin the decreased hemopoiesis, resulting in anemia [30].

## Limitation

In our study, we took secondary data for some patient characteristics. Besides, we did not assess the staple food of the study participants. Therefore, we admit that this cross-sectional study comes short of establishing specific causes of anemia in HIV-infected adults.

## Conclusion and recommendation

More than a third of HIV-infected adults in SGH are anemic. Therefore, it is a moderate public health problem in the area. In terms of severity, 90.6, 7.1, and 2.3% of the anemic patients have mild, moderate, and severe anemia, respectively. Having HIV infection for at least five years, a $CD_4$ cells count below 200 cells/mm$^3$, being on at least clinical stage III, and having a BMI below 18.5 kg/m$^2$ were found to be associated risk factors of anemia among HIV-infected adults in SGH.

Early enrolling of HIV-infected adults into ART with continuous monitoring of $CD_4$ cell count and good management initiated before the advancement of the clinical stage might reduce the observed high prevalence of anemia. Besides, nutritional support for HIV-infected

adults is important to reduce the risk of anemia among this particular population group in SGH.

## Supporting information

**S1 Dataset.**
(RAR)

## Acknowledgments

We thank Arba Minch University for allowing us to conduct this research. We thank study participants for taking part in this study. We also acknowledge the SGH staff for their cooperation and material support.

## Author Contributions

**Conceptualization:** Rishan Hadgu, Ahmed Husen, Esayas Milkiyas, Niguse Alemayoh, Robel Zemoy, Azene Tesfaye, Dagimawie Tadesse, Aklilu Alemayehu.

**Data curation:** Rishan Hadgu, Ahmed Husen, Esayas Milkiyas, Niguse Alemayoh, Robel Zemoy.

**Formal analysis:** Rishan Hadgu, Ahmed Husen, Esayas Milkiyas, Niguse Alemayoh, Robel Zemoy.

**Investigation:** Rishan Hadgu, Ahmed Husen, Esayas Milkiyas, Niguse Alemayoh, Robel Zemoy.

**Methodology:** Rishan Hadgu, Ahmed Husen, Esayas Milkiyas, Niguse Alemayoh, Robel Zemoy, Azene Tesfaye, Dagimawie Tadesse, Aklilu Alemayehu.

**Software:** Rishan Hadgu, Aklilu Alemayehu.

**Supervision:** Azene Tesfaye, Dagimawie Tadesse, Aklilu Alemayehu.

**Validation:** Rishan Hadgu, Ahmed Husen, Robel Zemoy, Azene Tesfaye, Dagimawie Tadesse, Aseer Manilal, Aklilu Alemayehu.

**Writing – original draft:** Rishan Hadgu, Dagimawie Tadesse, Aseer Manilal, Aklilu Alemayehu.

**Writing – review & editing:** Rishan Hadgu, Azene Tesfaye, Dagimawie Tadesse, Aseer Manilal, Aklilu Alemayehu.

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
