## [Decision Letter · Decision Letter 0]

29 Nov 2022

PONE-D-22-24145Prevalence, severity and associated risk factors of anemia among human immunodeficiency virus- -infected adults in Sawla General Hospital, southern Ethiopia: a facility-based cross-sectional study, 2019.PLOS ONE

Dear Dr. Alemayehu,

Thank you for submitting your manuscript to PLOS ONE. After careful consideration, we feel that it has merit but does not fully meet PLOS ONE’s publication criteria as it currently stands. Therefore, we invite you to submit a revised version of the manuscript that addresses the points raised during the review process.

We look forward to receiving your revised manuscript.

Kind regards,

Xinli Lu, Dr

Academic Editor

PLOS ONE

Journal Requirements:

2. "Previously Copyrighted Maps/Satellite images

We note that Figure 1 in your submission contain [map/satellite] images which may be copyrighted. All PLOS content is published under the Creative Commons Attribution License (CC BY 4.0), which means that the manuscript, images, and Supporting Information files will be freely available online, and any third party is permitted to access, download, copy, distribute, and use these materials in any way, even commercially, with proper attribution. For these reasons, we cannot publish previously copyrighted maps or satellite images created using proprietary data, such as Google software (Google Maps, Street View, and Earth). For more information, see our copyright guidelines: http://journals.plos.org/plosone/s/licenses-and-copyright.

Natural Earth (public domain): " ext-link-type="uri" xlink:type="simple">http://www.naturalearthdata.com/"

Reviewers' comments:

Reviewer's Responses to Questions

**Comments to the Author**

1. Is the manuscript technically sound, and do the data support the conclusions?

Reviewer #1: Partly

2. Has the statistical analysis been performed appropriately and rigorously? 

Reviewer #1: Yes

3. Have the authors made all data underlying the findings in their manuscript fully available?

Reviewer #1: Yes

4. Is the manuscript presented in an intelligible fashion and written in standard English?

Reviewer #1: No

5. Review Comments to the Author

Reviewer #1: I sincerely thank you for giving the opportunity to review this manuscript entitled as “Prevalence, severity and associated risk factors of anemia among human immunodeficiency virus- -infected adults in Sawla General Hospital, southern Ethiopia: a facility-based cross-sectional study, 2019.”. I hope this manuscript helps to understand the factors associated with anemia among PLWHIV. However, I have some comments that should be addressed by the authors of this manuscript.

1.In the variables section the authors stated that “Anemia was the dependent variable for this study. Age, sex, education, occupation, monthly income, residence, WHO clinical stage of HIV/AIDS, CD4 cell count, VL, body mass index (BMI), drug regimen type, duration of HIV infection, and HAART, were taken as dependent variables”. But the aforementioned variables are not dependent variable rather predictors or independent variables. In addition, I wonder if the authors of this manuscript have been measured the nutrition, disease related predictors of Anemia among PLWHIV. Since dietary intake is one of the known predictor of Anemia try to mention the common dietary practice in the study area, for instant what is the staple diet for most of the study population? Was the dietary data available? Is there any change in the dietary practice that might be differ from the other parts of Ethiopia? I advised the authors of this manuscript to clear out these things.

2.As indicated clearly in the manuscript you conducted a pretest. If there are any amendments/modifications resulted from pretest, please mention the result obtained

3.You stated that “Hosmer and Lemeshow's goodness of fitness for the logistic regression test was used to check model fitness” what was the result? Indicate the p-value

4.I also recommend to cite the references for the operational definitions

5.The result of the multivariate analysis also needs to be corrected “(AOR: 2.32; 95% CI: 1.08-4.94, P= 0.030); being on clinical stage III or more (AOR: 4.20; 95% CI: 1.06-16.62, P= 0.041); having CD4 count below 200 cells/mm3 (AOR: 4.32; 95% CI: 2.10-8.86, P= 0.000) and BMI below 18.5 kg/m2 (AOR: 3.82; 234 95% CI: 1.83-8.00, P= 0.000). As portrayed above the p-value for CD4 count and BMI was reported as 0.000, it should not be correct you may report as 0.001 or I strongly recommend to report only (AOR with 95%CI without reporting the p-value) since the aim of this study is to identify factors associated with anemia among PLWHIV. Apart from these please display only the result of the multivariate analysis in the table. Only the eight variables which have a p-value 0.2 logistic regression analysis and candidates of multivariate logistic regression analysis.

6.In the discussion section some of the reports were not completed for example Southwest Ethiopia _____ how much was the prevalence of anemia? Needs an intense revision and paraphrase to improve the readability of the manuscript…..I suggest to parphrase the following section; Various factors determine the risk of anemia among HIV-infected adults in multidimensional ways [3, 4]. In the current study, HIV-infected adults with CD4 cell count below 200 cells/mm3 (by264 4.32), BMI below 18.5 kg/m2 (by 3.82), clinical stage III or above (by 4.20), and having HIV infection for five or more years (2.32) have increased risk of anemia. Difficult to understand needs revision.

7.In general, this manuscript needs revision, i have some concerns about the grammar, and overall readability of the manuscript. Thus I will urge the authors of this manuscript to revise the text meticulously to fix the grammatical and typing errors and improve the overall readability of the text.

6. PLOS authors have the option to publish the peer review history of their article (what does this mean?). If published, this will include your full peer review and any attached files.

Reviewer #1: No

---

## [Author Response · Author response to Decision Letter 0]

13 Jan 2023

Rebuttal Letter

Dear editor and reviewers,

Greetings,

We have found your comments very important to improve our paper. Thank you very much. According to your comments, we have modified our paper. This rebuttal letter shows how we addressed your comments. For easy checkup of the document, we have put your comment with its corresponding response to editor’s comments followed by reviewer’s comments. Besides, we have uploaded the revised versions of our manuscript (one with tracked change and one without tracked change), and please kindly find them.

Editor’s comments and their corresponding responses

1. Conformance of our manuscript to PLOS ONE's style requirements

Response: Fully addressed by making modifications in the document formatting.

2. Copyright issues about location map of the study area.

Response: Fully addressed by removing the map.

3. Data availability issue

Response: Fully addressed by agreeing to make the data publicly available.

Reviewer’s comments and their corresponding responses

1. In the variables section the authors stated that “Anemia was the dependent variable for this study. Age, sex, education, occupation, monthly income, residence, WHO clinical stage of HIV/AIDS, CD4 cell count, VL, body mass index (BMI), drug regimen type, duration of HIV infection, and HAART, were taken as dependent variables”. But the aforementioned variables are not dependent variable rather predictors or independent variables.

Response: Fully accepted the comment. Based on the comment, the dependent and independent variables are now properly categorized.

In addition, I wonder if the authors of this manuscript have been measured the nutrition, disease related predictors of Anemia among PLWHIV. Since dietary intake is one of the known predictor of Anemia try to mention the common dietary practice in the study area, for instant what is the staple diet for most of the study population? Was the dietary data available? Is there any change in the dietary practice that might be differ from the other parts of Ethiopia? I advised the authors of this manuscript to clear out these things.

Response: Fully accepted the comment. Unfortunately, we did not assess such data. However, we accept this as an important comment for our future similar works.

2. As indicated clearly in the manuscript you conducted a pretest. If there are any amendments/modifications resulted from pretest, please mention the result obtained

Response: Fully addressed. Based on the pretest result, we did not made major modification to the questionnaire since we did not find major gap on it.

3. You stated that “Hosmer and Lemeshow's goodness of fitness for the logistic regression test was used to check model fitness” what was the result? Indicate the p-value

Response: Fully addressed. The model fitness was checked since our P value of Hosmer and Lemeshow's goodness of fitness was 0.05

4. I also recommend to cite the references for the operational definitions

Response: Fully addressed. Cited.

5. The result of the multivariate analysis also needs to be corrected “(AOR: 2.32; 95% CI: 1.08-4.94, P= 0.030); being on clinical stage III or more (AOR: 4.20; 95% CI: 1.06-16.62, P= 0.041); having CD4 count below 200 cells/mm3 (AOR: 4.32; 95% CI: 2.10-8.86, P= 0.000) and BMI below 18.5 kg/m2 (AOR: 3.82; 234 95% CI: 1.83-8.00, P= 0.000). As portrayed above the p-value for CD4 count and BMI was reported as 0.000, it should not be correct you may report as 0.001 or I strongly recommend to report only (AOR with 95%CI without reporting the p-value) since the aim of this study is to identify factors associated with anemia among PLWHIV. Apart from these please display only the result of the multivariate analysis in the table. Only the eight variables which have a p-value 0.2 logistic regression analysis and candidates of multivariate logistic regression analysis.

Response: Fully addressed. The sentences and numbers are corrected accordingly.

6. In the discussion section some of the reports were not completed for example Southwest Ethiopia _____ how much was the prevalence of anemia? Needs an intense revision and paraphrase to improve the readability of the manuscript…..I suggest to parphrase the following section; Various factors determine the risk of anemia among HIV-infected adults in multidimensional ways [3, 4]. In the current study, HIV-infected adults with CD4 cell count below 200 cells/mm3 (by264 4.32), BMI below 18.5 kg/m2 (by 3.82), clinical stage III or above (by 4.20), and having HIV infection for five or more years (2.32) have increased risk of anemia. Difficult to understand needs revision.

Response: Fully addressed. The sentences are corrected accordingly.

7. In general, this manuscript needs revision, i have some concerns about the grammar, and overall readability of the manuscript. Thus I will urge the authors of this manuscript to revise the text meticulously to fix the grammatical and typing errors and improve the overall readability of the text.

Response: Addressed and some grammatical corrections were made.

With kind regards,

Aklilu Alemayehu (Corresponding author)

---

## [Decision Letter · Decision Letter 1]

15 Mar 2023

PONE-D-22-24145R1Prevalence, severity and associated risk factors of anemia among human immunodeficiency virus-infected adults in Sawla General Hospital, southern Ethiopia: a facility-based cross-sectional study, 2019PLOS ONE

Dear Dr. Alemayehu,

Thank you for submitting your manuscript to PLOS ONE. After careful consideration, we feel that it has merit but does not fully meet PLOS ONE’s publication criteria as it currently stands. Therefore, we invite you to submit a revised version of the manuscript that addresses the points raised during the review process.

We look forward to receiving your revised manuscript.

Kind regards,

Frank T. Spradley

Academic Editor

PLOS ONE

Journal Requirements:

Reviewers' comments:

Reviewer's Responses to Questions

**Comments to the Author**

1. If the authors have adequately addressed your comments raised in a previous round of review and you feel that this manuscript is now acceptable for publication, you may indicate that here to bypass the “Comments to the Author” section, enter your conflict of interest statement in the “Confidential to Editor” section, and submit your "Accept" recommendation.

Reviewer #1: (No Response)

2. Is the manuscript technically sound, and do the data support the conclusions?

Reviewer #1: Partly

3. Has the statistical analysis been performed appropriately and rigorously? 

Reviewer #1: Yes

4. Have the authors made all data underlying the findings in their manuscript fully available?

Reviewer #1: Yes

5. Is the manuscript presented in an intelligible fashion and written in standard English?

Reviewer #1: Yes

6. Review Comments to the Author

Reviewer #1: I would like to say thank you for allowing me to review this manuscript entitled “Prevalence, severity and associated risk factors of anemia among human immunodeficiency virus-infected adults in Sawla General Hospital, southern Ethiopia: a facility-based cross-sectional study, 2019.”. I saw some improvements compared with the previous version of the manuscript. However, still this manuscript needs further revision.

1. Since it is secondary data you couldn’t find all the needed information, I understand that. But try to mention the common dietary practice of the population found in the study area. What is the staple diet of the study subjects? Is there any change in the dietary practice that might differ from the other parts of Ethiopia? Add some clarification ‘context” in the methods section to better understand the case

2. The result of table 3 is also not clear. You said that only variables that have an association at P 0.2 will be recruited to the final model. “multivariable logistic regression analysis was done for the variables with P value 0.20 in binary logistic regression analysis to identify risk factors associated with anemia among HIV-infected adults. Based on the cutoff point, eight variables were taken as candidates for the multivariable logistic regression analysis. Finally, anemia among HIV-infected adults was significantly associated with being HIV-infected for five or more years (AOR: 2.32; 95% CI: 1.08-4.94, P= 0.030); being on clinical stage III or more (AOR: 4.20; 95% CI: 1.06-16.62, P= 0.041), having CD4 count below 200 cells/mm346 (AOR: 4.32; 95% CI: 2.10-8.86, P 0.001) and BMI below 18.5kg/m2 47 (AOR: 3.82; 95% CI: 1.83-8.00, P 0.001)”

But in the table, you report 14 independent variables which have no any importance i.e. variables that have a p-value 0.2 in the univariate analysis. The purpose of variable selection in the multivariate analysis is to select clinically important and statistically significant variables while excluding unrelated variables so as to control the confounding factors. Hosmer and Lemeshow describe a purposeful selection of covariates within which an analyst makes a variable selection decision at each step of the modeling process…like Forward, Stepwise, and Backward etc. Hence, the authors of this manuscript try to fix the analysis method and report a final multivariate model which contains only 8 variables. Otherwise, clarify the case ….plausibly.

7. PLOS authors have the option to publish the peer review history of their article (what does this mean?). If published, this will include your full peer review and any attached files.

Reviewer #1: **Yes: **Mulualem Endeshaw

---

## [Author Response · Author response to Decision Letter 1]

18 Mar 2023

Rebuttal Letter

Dear editor and reviewer,

Greetings,

We have found your comments very important to improve our paper. Thank you very much. According to your comments, we have modified our paper. This rebuttal letter shows how we addressed your comments. For easy checkup, we have put the reviewer’s comment with its corresponding response. Besides, we have uploaded the revised versions of our manuscript (one with tracked change and one without tracked change), and please kindly find them.

Reviewer’s comments and their corresponding responses

1. Since it is secondary data you couldn’t find all the needed information, I understand that. But try to mention the common dietary practice of the population found in the study area. What is the staple diet of the study subjects? Is there any change in the dietary practice that might differ from the other parts of Ethiopia? Add some clarification ‘context” in the methods section to better understand the case

Response: Fully accepted the comment. Unfortunately, we did not assess such data. However, we accept this as an important comment for our future similar works. Furthermore, we have mentioned it under limitation part of this manuscript.

2. The result of table 3 is also not clear. You said that only variables that have an association at P 0.2 will be recruited to the final model. “Multivariable logistic regression analysis was done for the variables with P value 0.20 in binary logistic regression analysis to identify risk factors associated with anemia among HIV-infected adults. Based on the cutoff point, eight variables were taken as candidates for the multivariable logistic regression analysis. Finally, anemia among HIV-infected adults was significantly associated with being HIV-infected for five or more years (AOR: 2.32; 95% CI: 1.08-4.94, P= 0.030); being on clinical stage III or more (AOR: 4.20; 95% CI: 1.06-16.62, P= 0.041), having CD4 count below 200 cells/mm346 (AOR: 4.32; 95% CI: 2.10-8.86, P 0.001) and BMI below 18.5kg/m2 47 (AOR: 3.82; 95% CI: 1.83-8.00, P 0.001)”. But in the table, you report 14 independent variables which have no any importance i.e. variables that have a p-value 0.2 in the univariate analysis. The purpose of variable selection in the multivariate analysis is to select clinically important and statistically significant variables while excluding unrelated variables so as to control the confounding factors. Hosmer and Lemeshow describe a purposeful selection of covariates within which an analyst makes a variable selection decision at each step of the modeling process…like Forward, Stepwise, and Backward etc. Hence, the authors of this manuscript try to fix the analysis method and report a final multivariate model which contains only 8 variables. Otherwise, clarify the case ….plausibly

Response: Fully addressed. Based on the reviewer’s comment, we have removed the variables that were not included for multivariable analysis. Hence, the table included only eight variables.

With kind regards,

Aklilu Alemayehu (Corresponding author)

---

## [Editor Report · Decision Letter 2]

25 Apr 2023

Dear Dr. Alemayehu,

Thank you very much for submitting your manuscript to PLOS ONE, and for responding to our recent requests regarding your submission. PLOS ONE requires that research meets all applicable standards for the ethics of experimentation and research integrity (http://journals.plos.org/plosone/s/human-subjects-research). We reserve the right to reject any submission that does not meet our internal ethical standards, which in some cases are more stringent than local ethical standards.

Unfortunately, as you are not able to supply the original ethics approval document issued by Arba Minch University, we have concluded that your submission does not our ethical requirements for human subjects research submissions. We will therefore be overturning the provisional editorial accept decision, and will reject this manuscript.

I am very sorry that this issue was identified at such a late stage.

If in the future you are able to locate a copy of the original (signed/stamped) ethics approval document, you are welcome to submit the manuscript to PLOS ONE for further consideration. Please note that we might need to seek additional feedback from our editorial board on any resubmission before coming to a decision.

Kind regards,

Emily Chenette

Editor in Chief

PLOS ONE

---

## [Author Response · Author response to Decision Letter 2]

17 Jul 2023

As per the comment, we have attached the ethical clearance letter issued by Arba Minch University. Thank you for the comment.

---

## [Decision Letter · Decision Letter 3]

3 Nov 2023

PONE-D-22-24145R3

Prevalence, severity and associated risk factors of anemia among human immunodeficiency virus-infected adults in Sawla General Hospital, southern Ethiopia: a facility-based cross-sectional study, 2019

PLOS ONE

Dear Author(s),

Thank you for submitting your manuscript to PLOS ONE. After careful consideration, we feel that it has merit but does not fully meet PLOS ONE’s publication criteria as it currently stands. Therefore, we invite you to submit a revised version of the manuscript that addresses the points raised during the review process.

Please submit your revised manuscript supplemented with original ethics approval document issued by Arba Minch University by November 13, 2023. If you will need more time than this to complete your revisions, please reply to this message or contact the journal office at plosone@plos.org. Please include the following items when submitting your revised manuscript:

If applicable, we recommend that you deposit your laboratory protocols in protocols.io to enhance the reproducibility of your results. Protocols.io assigns your protocol its own identifier (DOI) so that it can be cited independently in the future. For instructions see: https://journals.plos.org/plosone/s/submission-guidelines#loc-laboratory-protocols. Additionally, PLOS ONE offers an option for publishing peer-reviewed Lab Protocol articles, which describe protocols hosted on protocols.io. Read more information on sharing protocols at https://plos.org/protocols?utm_medium=editorial-emailutm_source=authorlettersutm_campaign=protocols.

We look forward to receiving your revised manuscript.

Kind regards,

Mohammed Hasen Badeso, MPH in Field Epidemiology

Academic Editor

PLOS ONE

Journal Requirements:

Additional Editor Comments (if provided):

Dear Author(s),

Can you supply the original ethics approval document issued by Arba Minch University. Unless otherwise, we have concluded that your submission does not PLOS ONE ethical requirements for human subjects research submissions.

Reviewers' comments:

Reviewer's Responses to Questions

**Comments to the Author**

1. If the authors have adequately addressed your comments raised in a previous round of review and you feel that this manuscript is now acceptable for publication, you may indicate that here to bypass the “Comments to the Author” section, enter your conflict of interest statement in the “Confidential to Editor” section, and submit your "Accept" recommendation.

Reviewer #2: (No Response)

2. Is the manuscript technically sound, and do the data support the conclusions?

Reviewer #2: Yes

3. Has the statistical analysis been performed appropriately and rigorously? 

Reviewer #2: No

4. Have the authors made all data underlying the findings in their manuscript fully available?

Reviewer #2: No

5. Is the manuscript presented in an intelligible fashion and written in standard English?

Reviewer #2: Yes

6. Review Comments to the Author

Reviewer #2: General comments

1. In abstract it is better if you donot abbreviate terms (donot use Abriavation),You shall use full form

• HIV,SPSS,SGH,ART,BMI

2. In introduction (it is to wide please reduce to 1-1*1/2 page)

3. In methodology

• Study setting

o You shall abrievate sawla general hospital to SGH at line 97 b/c it is once abbreviated above

• Sampling technique put in sampling interval K value

• It is better if you put in figure format from total population to your sample with sampling interval

4. In data collection procedure it is better if you rewrite

o Clinical data as drug regimen, HIV stage, HAART exposure,duration of HIV

o Immunological variable as viral load and CD4

5. In operational definition pleas cite for definition HIV infected adults (15-64)

o Clearly explain frist line regimen and second line regimen you used for your research

6. In discussion it is better if you start with pathogenesis of HIV to cause anemia

7. In discussion and conclusion why you it is better to discuss for HIV infection duration, Since it have stastically significant association with anemia

7. PLOS authors have the option to publish the peer review history of their article (what does this mean?). If published, this will include your full peer review and any attached files.

Reviewer #2: **Yes: **

---

## [Author Response · Author response to Decision Letter 3]

5 Nov 2023

Rebuttal Letter

Dear editor and reviewer,

Greetings,

We have found your comments very important to improve the quality of our paper. Thank you very much for your consideration, time, comments and email. Accordingly, we have modified our manuscript following your comments. This rebuttal letter shows how we addressed your comments in detail. For easy checkup, we have put the reviewer’s comment (numbered and italicized) with its corresponding response (shaded by color). Besides, we have uploaded the revised version of our manuscript (one with tracked change and one without tracked change), and soft copy of the original ethical clearance letter written by IRB of AMU college of Medicine and Health Science (Reference number: IRB/0203196/11). Please kindly find the uploaded documents.

Reviewer’s comments and their corresponding responses

1. In abstract it is better if you donot abbreviate terms (donot use Abriavation), You shall use full form

• HIV, SPSS, SGH, ART, BMI

Response: Partially addressed the comment. We have replaced SGH and SPPSS with their expanded form. But, for the interest of space we have left the commonly known abbreviations as they are (i.e. HIV, AIDS, ART and BMI), hoping the esteemed reviewer will understand the condition.

2. In introduction (it is to wide please reduce to 1-1*1/2 page)

Response: Fully addressed. Based on the reviewer’s comment, we have summarized and shortened the introduction part.

3. In methodology

• Study setting

o You shall abrievate sawla general hospital to SGH at line 97 b/c it is once abbreviated above

Response: Fully addressed. Based on the reviewer’s comment, we have abbreviated it.

• Sampling technique put in sampling interval K value

Response: Fully addressed. Based on the calculation to determine sampling interval by considering source population and sample size, K=~ 4. Accordingly, we have described it inside the sampling technique section. Thank you!

• It is better if you put in figure format from total population to your sample with sampling interval

Response: Addressed. We have described the overall sampling technique in narration form including the sampling interval.

4. In data collection procedure it is better if you rewrite 

o Clinical data as drug regimen, HIV stage, HAART exposure,duration of HIV

o Immunological variable as viral load and CD4

Response: Fully addressed. We have independently categorized these variables into clinical and immunological thematic sub-category. Thank you for this important comment.

5. In operational definition pleas cite for definition HIV infected adults (15-64)

Response: Fully addressed. We have incorporated reference for operational definition of HIV-infected adults.

o Clearly explain frist line regimen and second line regimen you used for your research

Response: Fully addressed. We have incorporated reference for operational definition of HIV-infected adults.

6. In discussion it is better if you start with pathogenesis of HIV to cause anemia

Response: Fully addressed. Based on the reviewer’s comment, we have started the introductory paragraph of our discussion section with pathogenesis of anemia among HIV patients.

7. In discussion and conclusion why you it is better to discuss for HIV infection duration, Since it have stastically significant association with anemia.

Response: Fully addressed. Yes, duration of HIV infection is statistically significantly associated with anemia. Hence, it is already addressed in the discussion (paragraph 6) and conclusion (paragraph 1) sections. We thank the reviewer for the reminder.

Questions and their corresponding answers

1. In sample size determination why you did not use P value from study done Wolayita sodo rather than Debretabor? If you use p value from Wolayita sodo your sample size became 356 which provide large population than your study population?

Response: Fully addressed. The Wolaita Sodo paper was not published during our proposal development and data collection period (before October 2019). That paper was published on PLOS ONE in October 2019. Therefore, we have used the already published paper from Debre-Tabor Hospital. We thank the reviewer for this good comment to increase sample size and closer to our study area.

2. Why you do not included HIV infected adults who were infected after six month?

Response: Fully addressed. To allow chance for finding the effect (if it exists) of ART on anemia, VL, CD4 cell count and others, we have included adults who lived with HIV for at least six months. Besides, to increase the chance of finding CD4 and VL data since these tests are often done in referral linkage that will take time for sample transportation, testing, result receiving and registration.

3. Why you do not used data entry software (Epidata epinfo) for data entry which are better for increasing quality? b/c SPSS is data analysis software

Response: Fully addressed. Of course, we share the stand of our esteemed reviewer on the strength of epidata and epinfo to improve the quality of data during data entry. However, it is possible to directly enter into SPSS (if the data nature is less complex while taking great care to minimize error). We thank the reviewer for this critical comment and we will use these software in our future researches.

With kind regards,

Aklilu Alemayehu (Corresponding author)

---

## [Editor Report · Decision Letter 4]

13 Nov 2023

Prevalence, severity and associated risk factors of anemia among human immunodeficiency virus-infected adults in Sawla General Hospital, southern Ethiopia: a facility-based cross-sectional study

PONE-D-22-24145R4

Dear Author(s),

We’re pleased to inform you that your manuscript has been judged scientifically suitable for publication and will be formally accepted for publication once it meets all outstanding technical requirements.

Kind regards,

Mohammed Hasen Badeso, MPH in Field Epidemiology

Academic Editor

PLOS ONE
---

## [Editor Report · Acceptance letter]

1 Dec 2023

PONE-D-22-24145R4 

Prevalence, severity and associated risk factors of anemia among human immunodeficiency virus-infected adults in Sawla General Hospital, southern Ethiopia: a facility-based cross-sectional study 

Dear Dr. Alemayehu:

I'm pleased to inform you that your manuscript has been deemed suitable for publication in PLOS ONE. Congratulations! Your manuscript is now with our production department. 

Kind regards, 

on behalf of

Mr Mohammed Hasen Badeso 

Academic Editor

PLOS ONE